# COVID-19 Vaccine Hesitancy in Diverse Groups in the UK—Is the Driver Economic or Cultural in Student Populations

**DOI:** 10.3390/vaccines10040501

**Published:** 2022-03-24

**Authors:** Francis Drobniewski, Dian Kusuma, Agnieszka Broda, Enrique Castro-Sánchez, Raheelah Ahmad

**Affiliations:** 1Department of Infectious Diseases, Faculty of Medicine, Imperial College London, London W12 0NN, UK; a.broda@imperial.ac.uk; 2Centre for Health Economics & Policy Innovation, Imperial College Business School, London SW7 2AZ, UK; d.kusuma@imperial.ac.uk; 3College of Nursing, Midwifery and Healthcare University of West London, Middlesex, London TW8 9GB, UK; enrique.castrosanchez@uwl.ac.uk; 4Division of Health Services Research and Management, School of Health Sciences, City, University of London, London EC1V 0HB, UK; raheelah.ahmad@city.ac.uk

**Keywords:** COVID-19, vaccine hesitancy, students, healthcare workers

## Abstract

Studies have identified a greater reluctance for members of the Black, Asian, and minority ethnic communities to be vaccinated against COVID-19 despite a higher probability of greater harm from COVID-19. We conducted an anonymised questionnaire-based study of students (recruiting primarily before first reports of embolic events) at two London universities to identify whether economic or educational levels were primarily responsible for this reluctance: a postgraduate core group (PGCC) *n* = 860, and a pilot study of undergraduate medical and nursing students (*n* = 103). Asian and Black students were 2.0 and 3.2 times (PGCC) less likely to accept the COVID vaccine than White British students. Similar findings were noted in the pilot study students. As the students were studying for Master’s or PhD degrees and voluntarily paying high fees, educational and economic reasons were unlikely to be the underlying cause, and wider cultural reservations were more likely. Politicians exerted a strong negative influence, suggesting that campaigns should omit politicians.

## 1. Introduction

The behavioural responses of individuals and groups to the pandemic have been central to efforts to prevent and control viral transmission. Nonpharmaceutical interventions, including self- isolation, wearing face coverings and following lockdown rules and best practice guidance, have relied heavily on the public’s acceptance and sustained behaviour change. Now, with an established technological vaccine solution, there are additional behavioural responses required. First, the vaccine is one component of protection, and other prevention behaviours still need to be practised to reduce transmission. Second, and the focus of this paper, apart from the logistics of access, there is the individual decision to be made by each of us to take up the vaccine.

Across the globe, varying levels of uptake have been reported, and some controversial methods to increase uptake have been employed, from positive incentives (e.g., free sausages with vaccination in one German town, participation in lotteries in Hong Kong, Canada and the USA, direct cash in Serbia and Sweden) to sanctions for failure to be vaccinated (e.g., the government of Punjab in Pakistan has employed mobile phone SIM card blocking [1]. Several countries, including the UK, are considering mandatory vaccination for social and health care workers. The different approaches can be understood in terms of the hierarchical positions on the Nuffield ladder of interventions from ‘observe and monitor’ uptake all the way up to limiting choice and the possibility of regulation [2].

While we have sizeable parts of the population across the globe unvaccinated or partially vaccinated [3], every country is trying to identify the size and key determinants of those groups who hesitate over vaccine uptake in general and COVID-19 in particular. However, before we make the leap to ‘hesitancy’ or refusal, we must be sure that barriers to access have been addressed. For example, in the US, there are reports of protracted online booking systems, complex use of language, English only documentation, and refusal at centres due to lack of personal ID [4]. Opportunity costs quickly escalate for those groups already at a disadvantage—over a third of Black American households are without access to a computer or broadband, and one in five households lack access to a vehicle, relying solely on public transport [4]. With the backdrop of approximately 26.1 million individuals (8.1% of the U.S. population) without any health care insurance just before the pandemic began, and 55.4% relying on employer-provided coverage [5], this means the majority are in a highly vulnerable position should they lose employment. While the COVID vaccine is free in the US, irrespective of citizenship or immigration status, if your experience of USA health care has been negative due to economic reasons then this will influence knowledge, acceptance, and trust now. Why would an illegal migrant with limited language skills believe that COVID vaccination is free if nothing else is? In contrast, national health systems, free at the point of access, such as in the UK, address some of these barriers and forms of exclusion, at least from a health care perspective.

Nevertheless, in the UK, as in the USA, Black, Asian, and minority ethnic (BAME) groups are financially vulnerable to working in unstable employment; many live in higher density multigenerational households and are unable to work at home, making high-risk trade-offs between isolation and work, including higher use of public transport, contributing to increased risk.

Members of the BAME community have also been disproportionately affected by COVID-19, i.e., higher rates of infection, hospitalisation and death [6]. In the UK, multiple explanations have been offered for this, with poverty as a root underlying cause increasing the risk of transmission due to high household density in multigenerational households, zero-hours contracts prohibiting isolation, and work from home [7]. Fortunately, within a year of the identification and genomic sequencing of the viral cause of COVID-19, multiple highly protective vaccines have been developed. Countries such as the UK, Israel, Bahrain, and member states of the EU and the USA have rolled out highly successful vaccination programmes with significant proportions of the total adult populations covered.

In a UK survey in December 2020, vaccine hesitancy was highest among Black (odds ratio 12.96, 95% confidence interval 7.34 to 22.89), Bangladeshi, and Pakistani (both 2.31, 1.55 to 3.44) populations compared with people from a white ethnic background [8]. BAME health care workers have also shown hesitancy compared to their white co-workers [8]. Similarly, in the US, Black and Hispanic individuals were less willing than Whites to receive the COVID-19 vaccine [9,10].

Was this reluctance due to a lack of knowledge or understanding of vaccine efficacy or safety, underlying poverty preventing access and uptake, or deeper cultural reasons in the BAME community that is perhaps rooted in historical mistrust of state bodies including the health service?

Attempts to encourage vaccine uptake will depend on an understanding of the reasons underpinning the reluctance. We attempted to better understand this through our recent analysis of socioeconomic indicators, including gender, age, ethnicity, education, and being medical or nursing students.

## 2. Materials and Methods

A cohort of 860 postgraduate students completed an anonymised questionnaire relating to COVID vaccine hesitancy (questionnaire provided in Appendix A) at two leading universities in London. The postgraduate students (2150) who were working for a higher degree, including Master’s or PhD students, received a specific email with an access code to the questionnaire with a follow-up reminder. They were asked about their views before and after any reports of embolic side effects emerged [11]. In our analysis, we used February-March 2021 and April-May 2021 to identify before and after, respectively. The response rate was approximately 40% (those having been sent the email and completing the questionnaire), which was expected as the timing of the questionnaire was in the run-up to exams. In addition, a pilot study of 103 undergraduate medical and nursing students was conducted by posting information on relevant physical and virtual notice boards for medical and nursing students.

The survey tool was developed based on a review of constructs identified in existing literature reviews and primary studies for our specific target population (students/young people). The framework and questions were guided by the principles outlined in the development of a survey tool by the WHO Strategic Advisory Group of Experts on Immunization WHO SAGE Working Group [12]. The survey includes the following three constructs as they have been identified as the top three reasons for vaccine hesitancy reported in the WHO Global status of immunization safety report [13], namely: (a) beliefs, attitudes, motivation about health and prevention, (b) risk/benefit of vaccines (perceived risks, experiences (heuristics)), and (c) communication and media environment. Major issues were fear of side effects of vaccination and distrust in the vaccine, lack of perceived risk of vaccine-preventable diseases, and the influence of anti-vaccination reports in the media. Our survey is enhanced, as it also includes intention and behaviours regarding the influenza vaccine and is informed by the ‘five C’ scale to assess psychological antecedents of vaccination (Complacency, constraints, calculation, collective responsibility) [14]. The validation process included survey pre-test, revision, and pilot prior to implementation.

The main outcome variable is vaccine acceptance. For acceptance, participants responded affirmatively (agree/strongly agree) when asked “How do you feel about the COVID-19 vaccine today”? For uptake, participants responded yes when asked “Have you had a COVID-19 vaccination”? Moreover, we asked a series of questions related to levels of confidence in the vaccine, preferred conditions (e.g., I am more likely to take the COVID-19 vaccine if), sources of information about the vaccine, and the history of influenza vaccine. We also collected socioeconomic indicators, including gender, age, ethnicity, education, and being medical or nursing students.

At the time of questionnaire completion, the cohort would not have been of an age receiving routine vaccination in the UK, but many would have been vaccinated due to professional reasons, such as being a medical student in the hospital or a vaccine volunteer. We therefore included a question about COVID vaccination status.

We conducted descriptive and multivariate regression analysis. For descriptive analyses, we provided the sample characteristics and prevalence of participants who responded affirmatively (agree/strongly agree or yes). For regression analyses, we used multivariate logistic regression, controlling for socioeconomic variables. All analyses were conducted in STATA MP 15.1. We analysed the core postgraduate cohort (PGCC) as a uniform group and compared them with the pilot group of medical and nursing students where helpful.

Approval was obtained from the Imperial College Research Ethics Committee (Ref: 21IC6546) and City University Research Ethics Committee (Ref: ETH2021-0904). Informed consent was obtained from all participants.

## 3. Results

The demographic characteristics of the full cohort of students are included in Table 1 and show that students were predominantly between 22 and 30 years of age (Table 1).

Table 2 shows the level of confidence, preference, source of information, and flu vaccine history towards vaccine acceptance and uptake. For PGCC, 91% were confident that the COVID vaccines were safe (Panel a, Column 2). Belief in long-term safety was similar, as was the proportion who thought that the vaccine had been adequately tested. Overall, scientists and health care professionals had a strong positive influence on safety and efficacy perception, with an equally strong negative effect when statements were made by politicians. A small percentage (7%; Panel a, Row 9, Column 2) of all respondents preferred to “have COVID-19 and develop their own immunity.”

In general, individuals who were “vaccine hesitant” stated that they were more likely to take the COVID-19 vaccine if it were made available at the person’s place of work, if peer colleagues and hospital leaders had been vaccinated, and if there was an opportunity to ask questions about the vaccine (Panel b, Column 6).

Table 3 shows the associations between level of confidence, preference, source of information, flu vaccine history and vaccine acceptance and uptake. Having a previous influenza vaccine or current one was strongly indicative of a desire to have a COVID-19 vaccination. Those who had an influenza vaccine in any of the past three years were 6 times more likely to want the COVID- 19 vaccine (Panel d, Row 5, Column 1). A positive history of prior influenza vaccination (or view on the acceptability of influenza vaccination) provides a strong indicator of the likely acceptability of COVID-19 vaccination. This group of respondents would not have been routinely offered the influenza vaccine, as they were too young.

The majority, as expected, learned about COVID vaccination mainly from professional or scientific sources, but interestingly, with limited input from other media, including social media, despite the age profile of the group (Table 2, Panel d, Column 2).

Considering the correlates of vaccine acceptance (Table 4), older age was positively associated with vaccine acceptance both before and after revelations of embolic side effects of the AstraZeneca vaccine (which subsequently led to a non-AstraZeneca vaccine being chosen for younger age groups in the UK).

If one considers the entire cohort (i.e., the PG core plus the undergraduate medical and nursing students from the pilot study), similar trends were seen. Asian and Black students were 1.8× and 5× less likely to accept COVID vaccination compared to white British students in the total cohort, and were 2.0 and 3.2× less likely in the PG core cohort. Curiously, medical and nursing students were 1.92 and 3.06 times less willing to be vaccinated than other students. This willingness to be vaccinated needs to be viewed in the context of the findings that the medical and nursing students were 2.8 times more likely to have received the vaccine at the time of the survey. For the medical/nurse student group, it would appear that although there was a collective reluctance to be vaccinated, there was pragmatic acceptance.

## 4. Discussion

The key observation was that Asian and Black students were 2.0× and 3.2× LESS likely to accept the COVID vaccine compared to White British students. The same ethnic group findings were noted in those recruited before reports of embolisms [11] (up to 31 March 2021) and those, albeit a smaller sample, completing the questionnaire afterwards (up until 30 May).

We also explored the main sources of information on vaccine safety and efficacy in the study population, as this would be the key to influencing their views and opinions later on. It was clear that scientists/doctors had a strong positive influence on vaccine uptake, while politicians exerted a strong negative influence across all groups. Our findings strongly suggest that campaigns to increase vaccine confidence in BAME individuals in particular should therefore omit politicians.

In relation to the influenza vaccine, those who have had an influenza vaccine in any of the past three years were 6.5 times more likely to want the COVID vaccine compared to those who have not had an influenza vaccine. Influenza vaccination is a useful marker for COVID-19 vaccination, i.e., it indicates a generally supportive attitude to vaccination in general.

In this population group, knowledge of science, health and vaccines can be assumed to be high given that all participants have a bachelor’s degree and are studying for a master’s or PhD degree in health or medical sciences. We can rule out lack of knowledge/understanding as a major factor in vaccine hesitancy.

Although no direct questions were made regarding wealth, these postgraduate students voluntarily attended and paid for high-cost courses (a range of GBP 15,000 to over GBP 30,000). Within this group, we can conclude that the reasons some BAME groups are hesitant to be vaccinated cannot be due to lack of knowledge or because of poverty. Other factors, including deeply held cultural beliefs or social norms as well as prior experiences with health care or health care services, may be crucial determinants.

Our study conclusions are supported by those of Sturgis et al. (2021), who used pre-COVID cross-sectional pandemic data from the Welcome Global Monitor and showed that in countries where trust in science is high, people are also more confident about vaccination, accounting for their own level of trust in science. Countries where the consensus is that science and scientists can be trusted are high showed a positive association between that trust in science and vaccination confidence [12]. A more ethnically homogenous group of healthcare students in the US, however found lower hesitancy compared to the general population, although in fact the study did not collect data on ethnicity due to the low participant numbers, and so risk of identification in an otherwise anonymous survey [15].

The specific findings in our pilot study of medical and nursing students demonstrated similar findings, which would need verification through a larger study. However, this group did suggest that even trainee doctors and nurses would not automatically support COVID vaccination despite arguably being closer to the effects of the virus (patient deaths, largely greater work exposure). Worryingly, with 1.3 million NHS staff, this group may have a wider negative influence against vaccination amongst the general population as well.

If compulsory vaccination of NHS and social care staff is mandated (as originally proposed in the UK and subsequently cancelled), there is a risk of a negative impact on NHS staff recruitment and retention. Although the percentage staff lost would probably be small, this would be numerically significant in a workforce of the size of the NHS, adding to an existing shortfall of frontline clinical staff. If we accept that the policy is correct, then we must develop practical strategies that promote clinical staff retention against the policy background of compulsory vaccination. Table 5 gives a summary of factors that are likely to have a positive effect on COVID-19 vaccination but which would need to be verified in a larger cohort of NHS staff.

We accept that as the impact of COVID-19 may not be homogeneous across diverse ethnic groups, no single communication and engagement intervention may be effective in influencing behaviours in all communities. However, we identified positive (e.g., scientist) and negative influencers (e.g., politicians) for all groups. We believe this study will help to better tailor campaigns to increase vaccine uptake where needed and further inform existing initiatives aimed at all adults [6]. Close monitoring of uptake and learning for future campaigns will be essential to ensure that all ethnic groups are able and willing to be vaccinated. When low- and middle-income countries (LMICs) are unable to source sufficient vaccine doses despite great need, every behavioural strategy needs to be deployed to maximise uptake in countries which can afford more doses than their entire population. There may also be more similarities than differences between high-income and low-income settings in terms of behaviours and trusted sources; for example, a recent study shows that health workers are the most trusted sources of guidance about COVID-19 vaccines in LMICs [13].

Similarly, vaccine hesitancy during medical and nursing training should be addressed and arguably even beforehand during high school. As the UK faces complex decisions around release from lockdown and increasing case numbers, we need to consider the vaccination of teenagers (who carry and transmit but are largely immune to the lethal effects of the disease), and therefore family, student and teenager understanding and acceptance of vaccination both for individual health and for wider public health.

In terms of limitations, we have reported our approximate response rate, which is higher than comparable online surveys for similar groups [15]; this may have been a conservative measure, as we cannot be sure if all those sent the email opened and read the email invitation, especially as this was sent out through the central student communications office. Other studies have not necessarily stated their response rates [16]. Our study did not capture socioeconomic status, which might be a confounder within the medical and nursing groups.

## 5. Conclusions

These findings provide useful insight into disparities in uptake in future health care workers and provide opportunities for earlier interventions. For example, there may be implications for how we teach microbiology/infectious diseases literacy in our medical and nursing and other health-related courses. Understanding technology/vaccine development and safety may also be needed. There may be major implications as these students qualify and progress as health care professionals for vaccine uptake amongst the professional groups as well as the messages they relay to patients and the public at large. There has been much debate around the implementation of a mandatory vaccine policy for all those working in healthcare settings [17] more generally, there is learning of the relevance and acceptance of other intervention bundles and positive framing of activities encouraging vaccinations allowing activities (rather than lack of vaccination ‘preventing’ activities) such as the green pass initiative in similar economies and other population groups [18]. Future cross-country work would examine such intervention options across countries of different economies [19].

## Figures and Tables

**Table 1 vaccines-10-00501-t001:** Patient cohort sample characteristics.

		All Postgraduates (*n* = 860)		All Students (*n* = 963)	
		*n*	%	*n*	%
		[1]	[2]	[3]	[4]
(a) Characteristics					
Gender					
	Female	517	60.8	609	63.2
	Male	333	39.2	342	35.5
	Other	10	1.2	12	1.3
Age group					
	18–21	33	3.8	100	10.4
	22–24	313	36.4	327	34.0
	25–27	216	25.1	219	22.7
	28–30	110	12.8	113	11.7
	31–39	122	14.2	129	13.4
	40+	66	7.7	75	7.8
Ethnic					
	White	540	62.8	581	60.3
	Asian	198	23.0	232	24.1
	Black	47	5.5	60	6.2
	Others	75	8.7	90	9.4
Education					
	GCSE/A level	n/a	n/a	103	10.7
	Bachelor	329	38.3	329	34.2
	Master/PhD	520	60.5	520	54.0
	Other	11	1.3	11	1.1
Student med/nurse					
	Yes	106	12.3	177	18.4
	No	754	87.7	786	81.6
Education med/nurse					
	Yes	134	15.6	205	21.3
	No	726	84.4	758	78.7
(b) COVID-19 vaccine					
Vaccine acceptance					
	Yes	802	93.3	882	91.6
	No	32	3.7	52	5.4
	Undecided	26	3.0	29	3.0
Got vaccine (at least one dose)					
	Yes	252	29.3	311	32.3
	No	608	70.7	652	67.7
Among got vaccine, second dose					
	Yes	124	49.2	147	47.3
	No	128	50.8	164	52.7

Note: *n* = Observations.

**Table 2 vaccines-10-00501-t002:** Level of confidence, preference, source of information, flu vaccine history towards vaccine acceptance and uptake.

	Participants that Responded Affirmatively (Agree/Strongly Agree)									
	All Respondents		Vaccine Acceptance		Vaccine Hesitant		Got Vaccine		Not Yet Vaccine	
	*n* = 860		*n* = 802		*n* = 58		*n* = 252		*n* = 608	
	*n*	%	*n*	%	*n*	%	*n*	%	*n*	%
	[1]	[2]	[3]	[4]	[5]	[6]	[7]	[8]	[9]	[10]
(a) Levels of confidence in the vaccine										
I am confident that the COVID-19 vaccine available to me is safe	783	91%	773	96%	10	17%	239	95%	544	89%
2. I am confident about the safety of the first batch of vaccines developed	756	88%	749	93%	7	12%	235	93%	521	86%
3. I am confident about the long-term safety of the vaccine offered to me	703	82%	700	87%	3	5%	219	87%	484	80%
4. I am concerned about the immediate/short terms side effects of the vaccine	253	29%	226	28%	27	47%	69	27%	184	30%
5. I think that the risk of having the vaccine is greater than the risk of COVID-19	92	11%	70	9%	22	38%	26	10%	66	11%
6. I think the vaccine has been adequately tested	717	83%	707	88%	10	17%	220	87%	497	82%
7. I believe that the vaccine is not as good as it has been reported	94	11%	64	8%	30	52%	24	10%	70	12%
8. I think the vaccine would not work as well for me	25	3%	17	2%	8	14%	7	3%	18	3%
9. I would prefer to have COVID-19 and develop my own immunity	61	7%	40	5%	21	36%	18	7%	43	7%
10. I am unhappy that the second dose of vaccine is being delayed	464	54%	446	56%	18	31%	103	41%	361	59%
11. I do trust statements made about COVID-19 vaccine safety made by politicians	373	43%	368	46%	5	9%	105	42%	268	44%
12. I do trust statements made about COVID-19 vaccine safety made by scientists/doctors	796	93%	775	97%	21	36%	239	95%	557	92%
13. I do trust statements made about COVID-19 vaccine safety made by health care professionals (other than doctors)	716	83%	702	88%	14	24%	223	88%	493	81%
14. I do trust statements made about COVID-19 vaccine efficacy (how well the vaccine works) made by politicians	394	46%	389	49%	5	9%	111	44%	283	47%
15. I do trust statements made about COVID-19 vaccine efficacy (how well the vaccine works) made by scientists/doctors	805	94%	780	97%	25	43%	241	96%	564	93%
(b) I am more likely to take the Covid-19 vaccine if:	Participants that responded affirmatively (yes)									
Available at my place of work during working hours	261	88%	231	89%	30	77%	151	89%	110	86%
2. Available at my GP	265	87%	240	90%	25	68%	148	86%	117	89%
3. I am given time off from work afterwards	311	81%	280	81%	31	78%	125	71%	186	89%
4. I am updated on how many staff have had the vaccine	384	75%	356	75%	28	76%	106	52%	278	90%
5. Colleagues from the same profession have had the vaccine.	405	82%	374	82%	31	78%	125	67%	280	91%
6. Colleagues from different professions have had the vaccine	419	81%	389	81%	30	77%	127	65%	292	91%
7. Hospital leaders/management have had the vaccine	364	82%	333	82%	31	79%	134	71%	230	90%
8. I have an opportunity to ask questions and think about the vaccine before making a decision	369	91%	340	92%	29	85%	154	89%	215	93%
9. I have enough information about the safety of the vaccine	255	94%	230	96%	25	83%	155	96%	100	92%
10. Initial batches of vaccine have already been used successfully	270	90%	242	91%	28	82%	151	90%	119	91%
11. It was Recommended by my GP	332	85%	298	87%	34	69%	143	80%	189	89%
12. It was recommended by a scientific expert or doctor	249	91%	226	95%	23	64%	152	94%	97	87%
13. It was recommended by my religious leader, e.g., priest, Imam, rabbi, etc.	605	78%	569	79%	36	65%	113	48%	492	91%
14. It was recommended by a celebrity (e.g., TV or film star)	600	75%	563	76%	37	65%	99	42%	501	90%
15. It was recommended by someone famous from my age group	584	75%	548	76%	36	65%	96	41%	488	90%
(c) Sources of information about the vaccine-keeping up to date	Participants that responded affirmatively (yes)									
Official national sources	676	79%	631	79%	45	78%	205	81%	471	77%
2. Professional or scientific society	654	76%	608	76%	46	79%	203	81%	451	74%
3. Technical Sources/guidelines	576	67%	533	66%	43	74%	185	73%	391	64%
4. Professional network (online or in person)	480	56%	441	55%	39	67%	148	59%	332	55%
5. Social network (online or in person)	375	44%	342	43%	33	57%	102	40%	273	45%
6. Workers union	192	22%	177	22%	15	26%	60	24%	132	22%
7. Other Media formats	326	38%	296	37%	30	52%	92	37%	234	38%
(d) Out of the examples previously provided what was the principal source information about vaccines do you trust most?	Each participant chose one answer									
Professional or scientific society	538	63%	506	63%	32	55%	146	58%	392	64%
2. Official national sources	180	21%	174	22%	6	10%	61	24%	119	20%
3. Technical Sources/guidelines	95	11%	89	11%	6	10%	34	13%	61	10%
4. People, i.e., other health	13	2%	9	1%	4	7%	4	2%	9	1%
5. Other Media formats, i.e., Pharmaceutical	13	2%	9	1%	4	7%	2	1%	11	2%
6. Journalists and news	11	1%	10	1%	1	2%	2	1%	9	1%
7. Social media/Internet	8	1%	3	0%	5	9%	2	1%	6	1%
8. Organisation, i.e., Employer Workers union	2	0%	2	0%			1	0%	1	0%
(e) Did you have an influenza vaccine?	Participants that responded affirmatively (yes)									
Did you have an influenza vaccine?—Current winter (October 2020 till now)	206	24%	202	25%	4	7%	117	46%	89	15%
2. Did you have an influenza vaccine?—The last winter (October 2019–March 2020)	200	23%	194	24%	6	10%	108	43%	92	15%
3. Did you have an influenza vaccine?—The year before (October 2018–March 2019)	188	22%	183	23%	5	9%	89	35%	99	16%
4. Would you like to have an influenza vaccine this year?	270	44%	261	46%	9	18%	54	43%	216	44%
5. Did you have an influenza vaccine?—The past 3 years	304	35%	297	37%	7	12%	141	56%	163	27%

Note: *n* = Observation.

**Table 3 vaccines-10-00501-t003:** Associations between level of confidence, preference, source of information, influenza vaccine history and vaccine acceptance and uptake.

	Vaccine Acceptance		Got Vaccine	
	OR	SE	OR	SE
	[1]	[2]	[3]	[4]
**A. Levels of confidence in the vaccine (*n* = 860)**				
I am confident that the COVID-19 vaccine available to me is safe	210.25 **	(105.41)	3.69 **	(1.43)
2. I am confident about the safety of the first batch of vaccines developed	134.32 **	(66.72)	4.85 **	(1.74)
3. I am confident about the long-term safety of the vaccine offered to me	136.61 **	(86.27)	2.44 **	(0.63)
4. I am concerned about the immediate/short terms side effects of the vaccine	0.57	(0.17)	0.72	(0.15)
5. I think that the risk of having the vaccine is greater than the risk of COVID-19	0.15 **	(0.05)	0.7	(0.21)
6. I think the vaccine has been adequately tested	33.54 **	(12.83)	2.13 **	(0.57)
7. I believe that the vaccine is not as good as it has been reported	0.10 **	(0.03)	0.59	(0.19)
8. I think the vaccine would not work as well for me	0.20 **	(0.10)	0.6	(0.35)
9. I would prefer to have COVID-19 and develop my own immunity	0.12 **	(0.04)	0.81	(0.29)
10. I am unhappy that the second dose of vaccine is being delayed	2.84 **	(0.88)	0.57 **	(0.10)
11. I do trust statements made about COVID-19 vaccine safety made by politicians	8.13 **	(3.91)	1.11	(0.20)
12. I do trust statements made about COVID-19 vaccine safety made by scientists/doctors	51.85 **	(20.15)	3.62 **	(1.48)
13. I do trust statements made about COVID-19 vaccine safety by health care professionals (other than doctors)	19.75 **	(6.69)	3.05 **	(0.85)
14. I do trust statements made about COVID-19 vaccine efficacy (how well the vaccine works works) made by politicians	9.62 **	(4.69)	1.09	(0.19)
15. I do trust statements made about COVID-19 vaccine efficacy (how well the vaccine works worksworks) made by scientists/doctors	41.55 **	(16.04)	3.96 **	(1.80)
**B. I am more likely to take the COVID-19 vaccine if: (*n* = 860)**				
Available at my place of work during working hours	3.81 **	(1.89)	1.73	(0.71)
2. Available at my GP	4.59 **	(2.20)	1.12	(0.50)
3. I am given time off from work afterwards	1.95	(0.90)	0.51 **	(0.17)
4. I am updated on how many staff have had the vaccine	1.07	(0.50)	0.23 **	(0.06)
5. Colleagues from the same profession have had the vaccine.	1.52	(0.70)	0.33 **	(0.10)
6. Colleagues from different professions have had the vaccine	1.26	(0.58)	0.32 **	(0.09)
7. Hospital leaders/management have had the vaccine	1.43	(0.68)	0.42 **	(0.13)
8. I have an opportunity to ask questions and think about the vaccine before making a decision	2.19	(1.33)	1.72	(0.80)
9. I have enough information about the safety of the vaccine	3.61	(2.71)	3.7	(2.63)
10. Initial batches of vaccine have already been used successfully	2.18	(1.30)	0.91	(0.43)
11. It was Recommended by my GP	3.57 **	(1.54)	1.42	(0.53)
12. It was recommended by a scientific expert or doctor	16.99 **	(10.00)	8.33 **	(5.14)
13. It was recommended by my religious leader, e.g., priest, Imam, rabbi, etc.	1.74	(0.64)	0.19 **	(0.04)
14. It was recommended by a celebrity (e.g., TV or film star)	1.43	(0.54)	0.15 **	(0.03)
15. It was recommended by someone famous from my age group	1.36	(0.52)	0.14 **	(0.03)
**C. Sources of information about the vaccine-keeping up to date (*n* = 860)**				
Official national sources	0.88	(0.31)	1.36	(0.31)
2. Professional or scientific society	0.81	(0.29)	1.28	(0.28)
3. Technical Sources/guidelines	0.71	(0.23)	1.45	(0.28)
4. Professional network (online or in person)	0.63	(0.19)	1.15	(0.20)
5. Social network (online or in person)	0.54 **	(0.16)	1.23	(0.22)
6. Workers union	0.98	(0.33)	1.3	(0.28)
7. Other Media formats	0.54 **	(0.16)	1.09	(0.20)
**D. Did you have an influenza vaccine? (*n* = 860)**				
Did you have an influenza vaccine?—Current winter (October 2020 till now)	6.86 **	(3.96)	4.36 **	(0.88)
2. Did you have an influenza vaccine?—The last winter (October 2019–March 2020)	3.78 **	(1.83)	3.01 **	(0.60)
3. Did you have an influenza vaccine?—The year before (October 2018–March 2019)	4.55**	(2.38)	2.13 **	(0.44)
4. Would you like to have an influenza vaccine this year?	3.85 **	(1.53)	1.04	(0.24)
5. Did you have an influenza vaccine?—The past 3 years	6.00 **	(2.71)	2.63 **	(0.48)

Note: *n* = Observation, OR = Odds Ratios, SE = Standard errors. We ran a logit regression for each outcome variable. ** *p* < 0.05.

**Table 4 vaccines-10-00501-t004:** Sociodemographic correlates of vaccine acceptance (including before/after embolism issues) and uptake.

		Outcome: Vaccine Acceptance						Outcome: Got Vaccine	
		All Study Period		February–March 2021		April–May 2021			
		OR	SE	OR	SE	OR	SE	OR	SE
		[1]		[2]		[3]		[4]	
Gender									
	Female	Ref.							
	Male	1.41	(0.45)	1.36	(0.45)			0.55 ***	(0.11)
	Other	0.10 ***	(0.07)	0.13 **	(0.11)			0.43	(0.41)
Age group									
	18–21	Ref.							
	22–24	2.38	(1.39)	2.34	(1.58)	1.82	(2.80)	0.83	(0.39)
	25–27	2.34	(1.44)	2.14	(1.50)	5.65	(10.47)	1.37	(0.65)
	28–30	2.23	(1.50)	1.86	(1.38)			1.88	(0.95)
	31–39	1.95	(1.20)	1.71	(1.21)	3.98	(7.14)	4.08 ***	(1.96)
	40+	6.15 **	(4.92)	2.61	(2.42)	31.93 *	(61.88)	17.74 ***	(9.87)
Ethnicity									
	White	Ref.							
	Asian	0.50 **	(0.17)	0.48 **	(0.17)	0.85	(1.01)	0.91	(0.20)
	Black	0.31 **	(0.16)	0.32 *	(0.20)	0.19	(0.24)	1.42	(0.58)
	Other	0.52	(0.24)	0.49	(0.24)	-		0.68	(0.22)
Education									
	Bachelor	Ref.							
	Master/PhD	0.55 *	(0.19)	0.67	(0.26)	0.29	(0.28)	0.46 ***	(0.09)
	Other	0.21 **	(0.16)	0.18	(0.20)	0.08	(0.13)	0.43	(0.35)
Medical/nursing student									
	No	Ref.							
	Yes	0.52 *	(0.20)	0.55	(0.26)	0.72	(0.64)	3.06 ***	(0.75)
	Constant	14.10 ***	(8.57)	13.07 ***	(9.32)	11.11	(17.40)	0.38 **	(0.17)
	*n*	860		709		111		860	

Note: *n* = Observation, OR = Odds Ratios, SE = Standard errors. *** *p* < 0.01, ** *p* < 0.05, * *p* < 0.1.

**Table 5 vaccines-10-00501-t005:** Factors that should be incorporated in all health care and social care worker COVID-19 vaccination campaigns.

Recommendations and promotion made by scientists, doctors and health care workers No statements made by politicians Recommendations by GPs and religious leaders helpful Vaccine availability at place of work during normal working hours, i.e., minimal friction to maximise vaccine uptake Opportunity to ask questions regarding the vaccine Vaccine campaigns which build on influenza vaccine campaigns Consider positive incentives/rewards

## Data Availability

All data is provided in the manuscript.

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
