# Peer review of "COVID-19 Vaccine Hesitancy in Diverse Groups in the UK—Is the Driver Economic or Cultural in Student Populations"

_vaccines, 2022, doi:10.3390/vaccines10040501_

Round 1
Reviewer 1 Report
Thank you for giving me an opportunity to review this manuscript. The authors conducted a study based on anonymous questionnaire to identify whether economical or educational levels were primarily responsible for COVID-19 vaccine hesitancy. I agree this is an interesting and important research .But I have some comments as shown below:
1. In the line 42, a parenthesis is missing.
2. In the introduction part, the contents from the line 94 to the line 107 which are consistent with the material and methods part from the line 130 to the line 143. It would be better putting these contents in the material and methods part.
3. In the line 117, “the response rate was approximately 40%”, the quality control of questionnaire was not well, how to ensure the credibility of the results?
4. Probably the quality of the tables in table 3 could be improved in the last version.
5. This is a questionnaire-based study, what are the methods of sampling and identifying the sample? Please describe this in the method part.
6. The eleventh reference cannot be found in the article, please refer to the references in the correct place of the article.
Reviewer 2 Report
Thank you for asking me to review this article. The ongoing pandemic has resulted in global health, economic and social crises. Actually, the vaccination campaign is the first method to counteract the COVID-19 pandemic; however, sufficient vaccination coverage is conditioned by the people’s acceptance of these vaccines in the general population. In this context, the paper under review is aimed at identifing whether economic or educational levels were primarily responsible for COVID vaccine hesitancy in two London universities.
The subject under study is certainly important, especially in the historical period we are experiencing. The article presents interesting results but the manuscript must be improved especially for the local impact and the small sample of enrolled people. I would like to encourage authors to consider several issues to be improved.
Introduction: The authors should make clearer what is the gap in the literature that is filled with this study. The authors do not frame their study within the vast body of literature that addressed the issue of vaccine acceptance and vaccine hesitancy during the pandemic. What is the international situation regarding the acceptance of the vaccination in the adult population?
Methods: The survey was conducted using a non-standard questionnaire. The use of an unreliable instrument is a serious and irreversible limitation of the study. Moreover, no mention to a validation process is reported. What about face validity, reliability and intelligibility? The enrolment procedure must be better specified. How did the authors choose the way to enroll the sample? How did they avoid the selection bias? Why only two universities? What is the reference population? what is the minimum sample size?
Statistical analysis: I suggest to insert a measure of the magnitude of the effect for the comparisons. Please consider to include effect sizes.
Discussion: I also suggest expanding, emphasizing what is the possible international contribution of the study to the literature. What are the implications of the study? The discussion must be updated including the debated argument of a green pass linked to vaccination practice, if this issue was not considered by the author a paragraph should be added in the limit section with a proper reference (refer to articles with DOI: https://doi.org/10.3390/vaccines9111222).
Round 2
Reviewer 2 Report
The paper was improved and it is now suitable for publication